# Interleukin-6 and Macular Edema: A Review of Outcomes with Inhibition

**DOI:** 10.3390/ijms24054676

**Published:** 2023-02-28

**Authors:** Janine Yunfan Yang, David Goldberg, Lucia Sobrin

**Affiliations:** 1Department of Ophthalmology, Massachusetts Eye and Ear Infirmary, Harvard Medical School, Boston, MA 02114, USA; 2Eye Consultants of Pennsylvania, Reading, PA 19610, USA

**Keywords:** interleukin-6 inhibitors, tocilizumab, sarilumab, macular edema, non-infectious uveitis

## Abstract

This paper describes the current literature on the molecular pathophysiology of interleukin-6 (IL-6) in the genesis of macular edema and on the outcomes with IL-6 inhibitors in the treatment of non-infectious macular edema. The role of IL-6 in the development of macular edema has been well elucidated. IL-6 is produced by multiple cells of the innate immune system and leads to a higher likelihood of developing autoimmune inflammatory diseases, such as non-infectious uveitis, through a variety of mechanisms. These include increasing the helper T-cell population over the regulatory T-cell population and leading to the increased expression of inflammatory cytokines, such as tumor necrosis factor-alpha. In addition to being key in the generation of uveitis and subsequent macular edema through these inflammatory pathways, IL-6 also can lead to the development of macular edema through other pathways. IL-6 induces the production of vascular endothelial growth factor (VEGF) and facilitates vascular leakage by downregulating tight junction proteins in retinal endothelial cells. Clinically, the use of IL-6 inhibitors has been found to be efficacious primarily in the context of treatment-resistant non-infectious uveitis and secondary macular edema. IL-6 is a key cytokine in retinal inflammation and macular edema. It is thus not surprising that the use of IL-6 inhibitors in treatment-resistant macular edema in the setting of non-infectious uveitis has been well documented as an effective treatment option. The use of IL-6 inhibitors in macular edema secondary to non-uveitic processes has only begun to be explored.

## 1. Introduction

Interleukin-6 (IL-6) is a proinflammatory cytokine and a key player in multiple inflammatory cascades. It is produced systemically and intraocularly by monocytes, macrophages, and T cells and binds to a number of soluble and membrane-bound receptors [1,2,3] The effects of IL-6 include inducing the differentiation of CD-4-positive T-helper cells, augmenting the effect of transforming growth factor-beta, and inducing the production of vascular endothelial growth factor (VEGF) [1,2,3].

There are currently four different IL-6 inhibitors available from two different classes of medication: monoclonal receptor antibodies and monoclonal antibodies [1,2,3]. Monoclonal antibodies are antibodies produced by a single clone of B cells, resulting in monospecific and homogenous antibodies that allow targeted and therapeutic binding to a soluble antigen, such as IL-6. In contrast, monoclonal receptor antibodies are targeted antibodies to a specific receptor, such as the IL-6 receptor (IL-6R). Monoclonal receptor antibodies include tocilizumab, sarilumab, and satralizumab, while siltuximab is currently the only IL-6 monoclonal antibody biological drug available [1]. Tocilizumab is approved by the Federal Drug Administration (FDA) for use in treating uveitis in the setting of juvenile idiopathic arthritis (JIA), giant cell arteritis (GCA), rheumatoid arthritis (RA), and cytokine release syndrome [1,4]. Sarilumab is approved by the FDA for use in treatment-resistant RA [1,5]. However, in the setting of ocular disease, both tocilizumab and sarilumab are used off-label for treatment-resistant macular edema in the setting of uveitic and non-uveitic pathologies [2]. In particular, IL-6 inhibitors have been reserved for use in treatment-resistant macular edema unresponsive to traditional therapies such as steroids, tumor necrosis factor alpha (TNF-alpha) inhibitors, or anti-VEGF inhibitors [1,6]. 

The aim of this study is to provide a literature review on the current knowledge of IL-6 in the molecular pathophysiology of macular edema, as well as to summarize the results for IL-6 inhibitors in the treatment of macular edema in the setting of non-infectious uveitis (NIU) and non-uveitic diseases. A comprehensive literature search using PubMed and Google Scholar for studies published between 1990 and 2022 was performed using ‘tocilizumab’; ‘sarilumab’; or ‘interleukin-6′ and one of the following search terms: ‘macular edema’, ‘diabetic retinopathy’, ‘uveitis’, ‘macular degeneration’, ‘autoimmune retinopathy’, and ‘retinal vein occlusion’. These initial search criteria yielded 120 articles. Of these articles, 108 articles were excluded due to a lack of detailed information reporting treatment response specifically of macula edema, resulting in 12 selected articles. Based on the initial search queries, additional articles deemed relevant were also included in the literature review, including ‘pseudophakic edema’ and ‘acute retinal necrosis’, yielding one article on pseudophakic macular edema and two articles on acute retinal necrosis. All of these articles were included in the review. Basic science papers, case series, and clinical trials were all reviewed to find eligible papers that aligned with the goals of this study. Papers that elucidated the molecular pathophysiology of IL-6 in the genesis of macular edema were included. Studies that included analyses on the clinical effects of IL-6 inhibitors on cystoid macular edema (CME) as a primary or secondary endpoint were included. Studies that did not directly comment on the effect of IL-6 inhibitors on CME were excluded.

## 2. The Role of Interleukin-6 in Ocular Pathology

IL-6 is a protein encoded by a gene in chromosome 7p21 and functions via signal transduction after binding IL-6R [1,2,3]. The two major forms of IL-6R are the transmembrane receptor protein (IL-6R) and the soluble receptor (sIL-6R) [1,2,3]. In the classic pathway, the binding of IL-6 and IL-6R along with a transmembrane glycoprotein gp130 leads to intracellular signal transduction via activation of the Janus kinase, signal transducer, and activator of transcription 3 (STAT3) and JAK-SHP2-Ras-mitogen activated protein kinase (MAPK) pathways [1,2,3] (Figure 1). The pleiotropic effect of IL-6 is largely due to the range of cells expressing gp130, which allows pleiotropic and redundant signaling [1,2,3]. IL-6 has been shown to be a key player in protecting the host against environmental hazards by causing a signal cascade as a warning signal, with an acute increase in the IL-6 levels occurring immediately at the onset of an acute inflammatory event [1,2,3]. Due to its involvement in the protective response, IL-6 is produced by multiple cells of the innate immune system as part of the integrated defense system. Pathogenic stimulation via pathogen-associated molecular patterns (PAMPs) or non-infectious inflammatory stimulation via damage-associated molecular patterns (DAMPs) from injured cells leads to the activation of Toll-like receptors (TLR). The activation of TLR leads to the expression of inflammatory cytokines, including IL-6, TNF-alpha, and IL-1 beta (Figure 2) [1,2,3]. TNF-alpha and IL-1 beta may trigger a positive feedback loop with the additional expression of IL-6, allowing for a rapid increase in the IL-6 levels. Liver hepatocytes respond to IL-6 by producing acute-phase proteins, including C-reactive protein, serum amyloid A, haptoglobin, and fibrinogen, which are responsible for acute and chronic inflammatory processes [1,2,3]. IL-6 also induces B-cell and T-cell differentiation. IL-6 induces helper T-cell differentiation and suppresses regulatory T-cell differentiation [1,2,3]. By increasing the helper T-cell population over the regulatory T-cell population, IL-6 stimulation lowers the immunologic tolerance, resulting in a higher likelihood of developing autoimmune inflammatory disease [1,2,3].

IL-6 plays a role in immunogenicity and inflammation in both systemic and ocular immune-mediated diseases, including juvenile idiopathic arthritis and NIU. Several studies have measured elevated IL-6 levels in ocular fluids, including aqueous fluid or vitreous in patients with NIU [1,6]. 

Outside of uveitis, elevated levels of IL-6 have also been found in multiple ocular diseases, including central vein occlusion [1,6]. In addition to its role in inflammation, IL-6 also plays a role in the production VEGF, which results in angiogenesis and vascular permeability [1,6]. Two of the major causes of VEGF induction include IL-6 and hypoxia, both of which are present in ocular pathologies such as retinal vein occlusions [1,6]. Vitreous levels of IL-6 and VEGF have been correlated with the severity of ischemia in patients with ischemic central retinal vein occlusion (CRVO) and correlated with disease severity [1,6]. In a systematic review published by Minaker et al., multiple studies have shown a significant elevation of IL-6, IL-8, IL-10, and VEGF in aqueous and vitreous samples of patients with RVO compared to healthy controls [7].

In diabetic retinopathy, the significant elucidation of IL-6′s role has occurred. Long-term hyperglycemia-related oxidative stress and inflammation lead to diabetic retinal changes by increasing the vascular permeability and allowing fluid leakage into the retinal interstitium [8,9]. This blood–retinal barrier dysfunction and breakdown is a key part of pathogenesis in retinal deterioration, macular edema, and progressive vision loss [8,9]. IL-6 has been shown to play a role in decreasing the barrier function in retinal endothelial cells, allowing vascular leakage by downregulating tight junction proteins [8,9]. IL-6 normally mediates the recruitment of leukocytes by increasing Intercellular Adhesion Molecule 1 (ICAM-1) and Vascular Cell Adhesion Molecule 1 (VCAM-1) to assist adhesion to the vascular endothelium. In patients with diabetic retinopathy, elevated levels of ICAM-1 and VCAM-1 have been detected, likely due to increased IL-6 activation [8,9]. ICAM-1 and VCAM-1 are transmembrane proteins involved in the adhesion and migration of leukocytes and endothelial cells. In diabetic patients, increased ICAM-1 expression has been correlated to an increase in migrating neutrophils, allowing increased migration and perivascular infiltration, leading to inflammation and retinal edema [8,9]. In addition, elevated IL-6 and other cytokines have been demonstrated in proliferative diabetic retinopathy (PDR) and are associated with a stimulation of matrix metalloproteinase (MMP) production [1,6]. MMP are a family of proteinases that regulate tissue remodeling, inflammation, and injury [7] In particular, MMPs regulate the integrity of the blood–retinal barrier, activate inflammatory mediators, and assist in angiogenesis and neovascularization [10]. In a mouse model, the inhibition of IL-6 signaling was shown to reduce the diabetes-induced oxidative damage both systemically and within the retina [8,9].

Neovascular age-related macular degeneration (AMD) is a common cause of progressive vision loss in older patients. As with diabetic retinopathy, there is also significant evidence from in vitro and animal studies that IL-6 could play a significant role in the disease. There have been in vitro and in vivo studies published exploring the possible role of IL-6 in neovascular AMD disease. In an in vivo assay, elevated levels of IL-6 were found in mice after laser injury, with the main expression from macrophages, resulting in choroidal neovascularization and angiogenesis [11]. However, in IL-6 knockout mice, there was decreased choroidal neovascularization compared to the wild-type mice, indicating the role of IL-6 in stimulating choroidal angiogenesis [11]. Lastly, macrophages were found as the major IL-6R-positive cells present in the eye. An analysis of IL-6R-positive macrophages after laser injury revealed a transcriptional profile consistent with Signal Transducer and Activator of Transcription (STAT3) activation and angiogenesis, therefore confirming the relationship between IL-6 and STAT3 [12]. STAT3 activation has been shown to aid in immune cell recruitment and the promotion of choroidal neovascularization [12]. In a systematic review analyzing 16 studies on IL-6 levels in AMD patients, systemically elevated IL-6 levels have been correlated with late-stage neovascular AMD [12]. In addition, several studies have been published on aqueous humor cytokine levels in eyes with neovascular AMD compared to normal eyes after traumatic procedures such as intravitreal injection or cataract surgery [12]. Although these studies have found an increase in IL-6 levels in patients with neovascular AMD, the results were not statistically significant, and conclusions must be treated cautiously due to small sample sizes and imperfect control groups that also received injection or surgery [12]. As with the pathogenesis of diabetic retinopathy, the role of IL-6 in AMD is closely tied to the dysfunction of the endothelium, increased oxidative damage, and increased expression of VEGF, leading to angiogenesis and vascular proliferation [12].

## 3. Interleukin-6 Blockage in Non-Infectious Uveitis and Its Associated Macular Edema

### 3.1. Randomized Clinical Trials

The utility of IL-6 inhibitors for macular edema secondary to NIU has been studied for various forms of NIU. From our comprehensive literature search, twelve studies were selected, highlighting the role of IL-6 inhibitors specifically on non-infectious uveitic macular edema. Most of the studies were published focusing on the use of tocilizumab, with only one study published on sarilumab. The details of the study characteristics are summarized in Table 1.

TNF-alpha inhibitors are often regarded as first-line biologic therapy in NIU refractory to steroids and conventional immunosuppressive medications such as methotrexate, mycophenolate, and cyclosporine [6] However, when first-line therapies fail, IL-6 inhibitors are integral in treatment-resistant NIU, particularly in the setting of uveitis secondary to JIA. As one of tocilizumab’s on-label uses, its use in treating uveitis related to JIA has been well documented in multiple studies [2,13,14,15,16]. In the APTITUDE study, a multicenter phase 2 single-arm trial, 21 pediatric patients with JIA-associated uveitis previously refractory to methotrexate, and TNF-alpha inhibitors were treated with subcutaneous tocilizumab over six months [13]. Patients were dosed according to body weight, with patients over 30 kg receiving 162 mg every two weeks and patients under 30 kg given 162 mg every three weeks [13]. In the APTITUDE trial, the primary endpoint, a two-step or more decrease in level inflammation based on the Standardization of Uveitis Nomenclature (SUN) criteria, was not met, with only 34% of patients responding to treatment [13]. Despite the overall low clinical response rate, tocilizumab proved effective in completely resolving macular edema in three out of four patients (75%) [13]. The medication was well tolerated, with minimal adverse effects in less than one-third of patients; these included injection site reaction, arthralgia, and headache. Although tocilizumab treatment did not meet its treatment efficacy endpoint this study, there is still evidence that it may be useful in treating JIA-related uveitis refractory to prior immunomodulatory therapy or biologics, and it may be particularly efficacious in the treatment of refractory macular edema from JIA uveitis, as per the results of the APTITUDE study [13,15,17,18].

Another randomized clinical trial, the STOP-Uveitis trial, was published investigating the safety and efficacy of tocilizumab in the treatment of NIU [19]. The study was conducted over a period of six months in 37 patients treated with 4 mg/kg or 8 mg/kg monthly intravenous infusions of tocilizumab [19]. A majority of patients had idiopathic panuveitis, with a minority of patients diagnosed with sarcoidosis, Vogt-Koyanagi-Harada syndrome, or punctate inner choroidopathy. At the baseline, 40.5% of patients were diagnosed with macular edema, with an average central foveal thickness (CFT) of 497 μm. The mean change in CFT after six months was −131.5 μm in the group treated with 4 mg/kg and −39.92 μm in the group treated with 8 mg/kg without any statistically significant difference between the two treatment groups. In addition to improvement in macular edema, patients also demonstrated improvement in best corrected visual acuity (VA) and inflammation measured as vitreous haze scores. No differences were found in treatment outcomes in the two treatment doses, although no conclusions may be drawn about ideal dosage due to the small sample size and exploratory nature of the trial.

Sarilumab is less commonly used in the treatment of ocular disease. Sarilumab is a human anti-IL-6R monoclonal antibody blocking the same classic and trans-signaling pathways as tocilizumab [6]. The SATURN study was published in 2018 investigating the effects of sarilumab in the treatment of NIU [20,21]. This randomized controlled study recruited 58 eyes with NIU. Participants were divided into the treatment group, which received biweekly subcutaneous sarilumab 200 mg over the course of four months, and a group that received a placebo [20,21]. The trial showed a beneficial effect on macular edema as measured by optical coherence tomography (OCT), with corresponding changes in the CFT measured as −46.8 μm in the treatment group versus +2.6 μm in the placebo group [20,21]. In a subgroup analysis of patients with baseline CFT greater than 300 μm, the overall change was more dramatic, with −112.5 μm in the treatment group versus −1.8 μm in the placebo group [20,21]. This difference in change of CFT was found to be statistically significant in the cohort overall, although the difference was not significant in the subgroup analysis, likely due to the small sample size. Overall, sarilumab treatment resulted in good clinical response measured by a two or more step reduction in vitreous haze on the Miami scale or reduction of the steroid usage, with a clinical response rate of 46% versus 30% in the treatment and placebo groups, respectively [20,21]. Sarilumab was well tolerated with minimal reported adverse events that were clinically insignificant compared to the placebo group. This study demonstrates the utility of sarilumab in treatment of NIU, particularly in uveitic macular edema, and warrants future studies with larger cohorts to corroborate the results.

**Table 1 ijms-24-04676-t001:** Literature review of studies of patients with refractory uveitis-related cystoid macular edema treated with IL-6 inhibitors.

First Author	Medication (Administration)	Year	Study Type	Study Objective	# Patients	Type of Uveitis	Follow up Time (Months)	Primary Outcome Measure	% of Primary Efficacy	% of Improvement in ME *	Definition of Improvement in ME *
Leclercq [17]	TCZ (IV)	2021	CS	Efficacy of TNF-alpha agents or TCZ in treating refractory uveitic macular edema	204	Idiopathic (97), BD (35), BCR (23), Sarcoidosis (15), JIA (12), VKH (8), spondyloarthritis (5), other (9)	6	Composite clinical response (CFT < 300 micrometer, RSU)	36%	56% (31/55)	CFT < 300, resolution of cystic spaces
Ramanan [8]	TCZ (SUBQ)	2020	SAT	Efficacy of TCZ in treating JIA associated NIU refractory to TNF-alpha and MTX	21	JIA	9	IOI	34%	75% (3/4)	N/A
Wennink [18]	TCZ (IV)	2020	CS	Efficacy of TCZ in treating NIU	7	Idiopathic uveitis (5), panuveitis (2)	18	CFT	100%	100% (7/7)	N/A
Vegas-Revenga [6]	TCZ (IV)	2019	CS	Efficacy of TCZ in treating refractory uveitic macular edema	25	JIA (9), BD (7), BCR (4), idiopathic panuveitis (4), sarcoidosis (1)	12	CFT	100%	100% (25/25)	CFT < 300, resolution of cystic spaces
Ozturk [19]	CS (IV)	2018	CS	Efficacy of TCZ in refractory BD	5	BD	10	CFT	100%	80% (4/5)	N/A
Heissigerova [16]	SAR (SUBQ)	2018	RCT	Efficacy of SAR in treating NIU (SAR vs. placebo)	58	Intermediate uveitis (12), posterior uveitis (14), panuveitis (29)	4	RSU	46%	100% (58/58)	20% reduction of CFT from baseline
Mesquida [20]	TCZ (IV)	2018	CS	Efficacy of TCZ in treating refractory uveitic macular edema	12	JIA (6), BCR (2), idiopathic panuveitis (2), sympathetic ophthalmia (1), ankylosing spondylitis (1)	24	CFT	100%	83% (10/12)	CFT < 350, resolution of cystic spaces
Sepah [14]	TCZ (IV)	2017	RCT	Efficacy of two different doses of TCZ in NIU (4 mg/kg versus 8 mg/kg)	37	Idiopathic (28), Sarcoidosis (2), VKH (2), BCR (2), PIC (1), BD (1), TINU (1)	6	IOI	44%	100% (15/15)	N/A
Tappeiner [10]	TCZ (IV)	2016	CS	Efficacy of TCZ in treating NIU	5	JIA	12	IOI	100%	100% (5/5)	N/A
Deuter [21]	TCZ (IV)	2016	CS	Efficacy of TCZ in treating refractory uveitic macular edema	5	JIA (2), AS (1), RA (2)	14	CFT	75%	100% (5/5)	25% reduction of CFT from baseline
Mesquida [20]	TCZ (IV)	2014	CS	Efficacy of TCZ in treating refractory uveitic macular edema	7	BCR (3), JIA (3), idiopathic panuveitis (1)	15	CFT	100%	71% (5/7)	CFT < 350
Adan [12]	TCZ (IV)	2013	CS	Efficacy of TCZ in treating refractory uveitic macular edema	5	BCR (3), JIA (1), idiopathic panuveitis (1)	8	CFT	100%	100% (5/5)	CFT < 350

* Improvement in macular edema was defined according to the study guidelines, or if undefined, criteria of the final CFT less than 350 micrometers were imposed. Table abbreviations: ankylosing spondylitis (AS), Bechet’s disease (BD), birdshot chorioretinopathy (BCR), central foveal thickness (CFT), case report (CR), case series (CS), intraocular inflammation (IOI), juvenile idiopathic arthritis (JIA), macular edema (ME), non-infectious uveitis (NIU), punctate inner choroidopathy (PIC), randomized controlled trial (RCT), reduced steroid use (RSU), single arm trial (SAT), tubulointerstitial nephritis and uveitis syndrome (TINU), Vogt-Koyanagi-Harada disease (VKH), subcutaneous drug administration (SUBQ), and intravenous drug administration (IV).

### 3.2. Comparative Retrospective Study

One multicenter retrospective observational study by Leclercq et al. sought to identify differences in clinical response in refractory uveitic macular edema, comparing tocilizumab versus anti-TNF-alpha treatment [22]. Two hundred and forty patients with prior treatment failure with traditional immunomodulatory therapy were included in the study with the biologic treatment chosen by the physician, resulting in 149 patients treated with infliximab or adalimumab and 55 patients treated with tocilizumab [22]. An improvement in CME was defined as any reduction in the baseline CFT with a lack of intraocular inflammation and less than 10 mg of corticosteroids per day. A partial response was defined as any improvement in CME, while a complete response was defined as the absence of cystic spaces on imaging and CFT measuring less than 300 μm. Overall improvement in macular edema was achieved in 46.2% of patients treated with anti-TNF alpha agents and 58.5% of patients treated with tocilizumab [22]. Complete or partial responses of macular edema were achieved in 21.8% of patients treated with anti-TNF alpha agents versus 35.8% of patients treated with tocilizumab [22]. In addition, tocilizumab improved the odds of a complete response of uveitic macular edema by more than two-fold in comparison to anti-TNF alpha agents after six months [22]. While 82 patients also received concomitant immunosuppressive therapy, there were no significant differences in type and distribution of concomitant treatment across the two treatment groups [22]. This study concluded that tocilizumab may be more effective than TNF-alpha inhibitors in treating treatment-resistant uveitic macular edema, although the findings should be confirmed ideally in a large prospective study.

### 3.3. Retrospective Case Series

Multiple case series have been published documenting the use of tocilizumab in treating NIU secondary to multiple etiologies, including JIA, birdshot chorioretinopathy, Bechet’s disease, and idiopathic panuveitis. The measurement of a clinical response as a primary endpoint varied by study, with most studies focusing on CFT or intraocular inflammation (IOI) as a marker for improvement. For this review, both the primary endpoint of the study and improvement in macular edema measured as a reduction in CFT (even if it was not the primary endpoint of the study) were analyzed. The primary endpoint was most commonly defined as improvement in macular edema measured by OCT features and/or CFT measurement or improvement in IOI measured by VA or vitreous haze score, as indicated in Table 1. 

While the primary endpoint was not met in at least 50% of patients in one study, the majority of patients demonstrated improvement in macular edema compared to the baseline [22]. In the three studies that did not specify criteria for improvement in macular edema, an average of 88% of patients demonstrated an objective decrease in macular edema based on CFT measurements [15,23,24]. The remaining five studies with criteria for improvement in macular edema had different definitions for this improvement. Leclercq et al. and Vegas-Revenga et al. defined improvement as reduction of CFT to under 300 μm, with 56% and 100% of patients demonstrating improved macular edema, respectively [6,22]. Mesquida et al. and Adán et al. defined improvement as reduction of CFT to under 350 μm, with 100% of patients demonstrating improved macular edema in both studies [25]. Deuter et al. defined improvement as a reduction of CFT of at least 25% of baseline thickness, with 100% of patients demonstrating improved macular edema [26]. Overall, these nine selected retrospective case series demonstrated a positive response to tocilizumab treatment in treatment-resistant uveitic macular edema.

### 3.4. Adverse Events

Overall, of the twelve highlighted studies, adverse events were reported in one study on sarilumab and six studies on tocilizumab. Adverse effects with tocilizumab included injection site reaction, arthralgia, headache, nausea, grade one leukopenia, and community acquired pneumonia [6,13,19,22,24,25]. Adverse effects with sarilumab included worsening uveitis and retinal infiltrates [21]. These two medications are generally well tolerated with mild to moderate severity levels of adverse effects.

### 3.5. Limitations of Existing Data

While all the selected studies have shown positive treatment response to tocilizumab in NIU-related macular edema, the findings are not directly comparable due to differences in study design, primary outcome measurements, patient clinical history, and definitions of improvement. Each study included different etiologies of NIU, and the distribution of anatomic location and etiologies were not consistent. The differences in treatment response may be due to differences in the various etiologies of NIU. The availability of detailed data on the clinical response of macular edema is limited based on the primary outcome of the study. In five studies, the primary patient population was focused on NIU with a smaller subgroup of patients with NIU-related macular edema [13,19,22,23,24]. Therefore, detailed information on defining improvement or resolution in macular edema based on CFT measurement or OCT findings were limited. In addition, criteria for improvement in macular edema varied by study including a maximum CFT threshold value regardless of baseline CFT, a minimum percentage CFT reduction from baseline CFT, and resolution of cystic spaces on OCT. As none of the studies defined criteria for macular edema at baseline, it is difficult to discern and compare the scale of impact on reduction of macular edema between studies. 

Another thing to consider in generalizing study results includes prior or concomitant immunomodulatory therapy. Many studies included patients who were receiving concomitant immunomodulatory therapy during the study period with tocilizumab. Vegas-Revenga et al. evaluated treatment-resistant macular edema with 12 out of 25 patients receiving concomitant methotrexate or cyclosporine [6]. Mesquida et al. evaluated treatment-resistant macular edema with eight out of twelve patients on concomitant oral prednisone, two patients on concomitant cyclosporine and one patient on concomitant methotrexate [25]. Adán et al. evaluated treatment-resistant NIU with seven out of eight patients on concomitant oral prednisone treatment [17]. Ozturk et al. evaluated treatment-resistant Bechet’s disease with all five patients receiving different concomitant immunomodulatory therapies [24]. Deuter et al. evaluated treatment-resistant macular edema with four out of five patients continuing concomitant immunomodulatory therapy and all five patients continuing oral prednisone during tocilizumab treatment [26]. Although all studies focused on patients with NIU or NIU-related macular edema with prior treatment failure, differences in concomitant therapy must be considered when evaluating the study results.

In conclusion, multiple studies have been published documenting the use of IL-6 inhibitors in treatment of both NIU and NIU-related macular edema. The twelve selected studies exhibited positive treatment response in improvement or resolution of macular edema. Of the twelve studies, all studies demonstrated multiple patients with a positive clinical response in reduction or complete resolution of uveitic macular edema by the end of the study period. Although not all patients responded to IL-6 therapy, a subset of treatment-refractory patients may benefit from its use. Overall, IL-6 inhibitors should be considered as a useful secondary therapy when traditional treatments are ineffective either as monotherapy or in conjunction with concomitant immunomodulatory therapy in the treatment of NIU and NIU-related macular edema. These results should be investigated in larger studies and clinical trials.

## 4. Interleukin-6 Blockage in Other Etiologies of Macular Edema

### 4.1. In the Setting of Acute Retinal Necrosis

Acute retinal necrosis (ARN) is an ocular infection caused by herpetic viruses causing complications, including retinal detachment, optic nerve atrophy, neovascularization, and cystoid macular edema [27]. The pathogenesis of edema is thought to be the destruction of the inner and outer blood–retinal barrier, leading to an accumulation of intraretinal fluid [27]. Refractory macular edema after ARN may be treated with intravitreal anti-VEGF injections or interferon alpha-2a [27]. However Bograd et al. presented a patient with recurrent severe CME after acute retinal necrosis [27]. This pediatric patient initially presented with painless blurry vision and redness of the right eye. Initial examination revealed anterior chamber inflammation, vitreous haze, and multiple areas of necrotic retinochoroiditis most consistent with ARN. There was no evidence of macular edema at first presentation. The diagnosis was confirmed with positive serology and positive anterior chamber tap for varicella zoster virus, and the patient was started on intravenous acyclovir, oral prednisone and topical prednisolone drops. After three months, active retinitis had subsided. However, the patient still had persistent inflammation on examination and newfound CME diagnosed on OCT. Despite treatment with oral prednisone and multiple intravitreal ranibizumab injection, the cystoid macular edema would recur after injections without complete resolution. After eleven months of treatment, the patient began receiving intravenous tocilizumab 8 mg/kg every four weeks, as well as being switched to aflibercept injections every four weeks [27]. After three months of treatment, the CME had improved significantly. After two years of continuous therapy, no evidence of inflammation was found [27]. By combining two treatments targeting different mediators of inflammatory CME secondary to ARN, this treatment-refractory patient demonstrated good clinical response. 

### 4.2. In the Setting of Non-Uveitic Macular Edema

IL-6 inhibitors have also demonstrated their use in treating treatment-refractory macular edema due to post-operative complications or retinal insults. Post-operative CME is a common complication of cataract surgery due to the disruption of the blood–retinal barrier from inflammatory mediators due to surgical manipulation [28]. While most acute cases resolve, chronic CME may lead to decreased vision and permanent distortion of photoreceptor architecture. Pham et al. described a case of a patient with worsening vision in the right eye for one month [28]. Past ocular history was significant for bilateral cataract extraction and posterior chamber intraocular and YAG laser capsulotomy. Slit lamp examination was normal without evidence of inflammation. Fluorescein angiography revealed perifoveal leakage without macular or retinal ischemia in the right eye, and OCT revealed presence of intraretinal and subretinal fluid in the right eye with mild epiretinal membranes in both eyes. Laboratory work-up for uveitis and malignancy was negative. The patient was diagnosed with persistent CME and started on topical prednisolone 1% without improvement and later started on oral prednisone. Despite systemic steroid treatment and topical interferon alpha 2b treatment, the CME progressed over four months. After receiving monthly infusions of 8 mg/kg tocilizumab for four months, OCT demonstrated complete resolution of CME [28]. This case represents the benefits of IL-6 inhibitors in treatment-resistant postoperative CME.

Non-paraneoplastic autoimmune retinopathy (npAIR) is a poorly understood retinopathy characterized by anti-retinal antibodies, visual field deficits, and global photoreceptor dysfunction with progressive vision loss [29,30,31]. CME, a complication of npAIR, is treated with topical steroids, topical non-steroidal anti-inflammatory drugs (NSAIDs), and intravitreal steroid injections [29,30,31]. However, the persistence of CME leads to escalation of care to immunomodulatory therapy and biologic drugs. There have been two published case reports and one case series with eight patients on the use of IL-6 inhibitors in the setting of treatment-resistant CME due to npAIR [29,30,31]. In all three studies, patients initially presented with unexplained progressive vision loss either unilaterally or bilaterally, leading to comprehensive multimodal imaging and anti-retinal antibody serology confirming the diagnosis of npAIR. Due to the initial presentation with CME diagnosed on OCT, all patients were started on topical steroids and NSAIDs, with three patients escalated to intravitreal steroid injections. Eight out of the ten patients later transitioned to immunosuppressive therapy with mycophenolate mofetil or azathioprine. Due to the persistent macular edema, two out of the ten patients began initial biologic therapy with rituximab or adalimumab without improvement. Due to a persistent lack of treatment response, all patients were initiated on intravenous tocilizumab or subcutaneous sarilumab monotherapy. Evidence of improvement in cystoid macular edema was observed with complete resolution after one to six months. One case report additionally demonstrated slight improvement in electroretinography amplitudes and reduction of leakage on fluorescein angiography after six months of sarilumab injections [30]. In the case series, all patients demonstrated ellipsoid zone recovery over 18 months [29]. Although the treatment of npAIR is difficult without clear guidelines, the utility of IL-6 inhibitors in treating refractory CME should be explored.

### 4.3. In the Setting of Retinal Vascular Pathology

Although the use of IL-6 inhibitors has been demonstrated in CME linked to inflammatory processes, both uveitic and non-uveitic, their use in retinal vascular pathologies has yet to be clinically explored. In common retinal vascular diseases such as diabetic retinopathy, neovascular AMD, CRVO, and branch retinal vein occlusion (BRVO), the mainstay of treatment has been anti-VEGF injections [8,9]. However, some patients demonstrate persistent macular edema despite anti-VEGF injections. IL-6 blockade could be an additional treatment option in these patients.

Retinal vein occlusion (RVO) is the second-most common cause of vision loss from retinal vascular disease [32,33]. Chronic complications include macular edema, neovascularization, and macular ischemia. The pathogenesis of RVO is thought to be related to venous obstruction, causing elevated venous pressure eventually contributing to overloading the collateral drainage system [7,33]. Macular edema secondary to RVO arises from efflux of fluid from the affected vessels due to Starling’s law due to elevated intravenous pressure, leading to the breakdown of the blood–retinal barrier and secretion of VEGF [33]. Currently, the management of RVO is focused on optimizing the systemic risk factors and treating complications such as neovascularization and macular edema [32,33]. The role of inflammatory markers such as VEGF and IL-6 has been explored in multiple studies. However, no clinical studies of IL-6 inhibition in humans with RVO have been performed to date.

The exact role that IL-6 plays in the treatment of diabetic retinopathy and diabetic macular edema still needs to be explored further in patients as well. A phase 2 clinical trial titled READ-4 (Ranibizumab for Edema of the mAcula in Diabetes: Protocol 4 with Tocilizumab) was proposed in 2017 to investigate the effects of tocilizumab intravenous infusions alone, tocilizumab combined with intravitreal ranibizumab injections, and ranibizumab injections alone on disease progression [34]. Unfortunately, the trial was discontinued in 2018 due to lack of funding. Although there are no formally published case series or trials on the use of IL-6 inhibitors in the treatment of diabetic retinopathy, this remains a promising area for potential exploration.

Similar to RVO and diabetic retinopathy, there are no published trials of IL-6 inhibition in the treatment of neovascular AMD. Human studies are needed to understand if anti-IL-6 treatment is a viable treatment option in AMD.

As described above, there are many similarities in the disease pathogenesis among vascular retinal diseases, most notably the initiation of damage and inflammation, leading to an inflammatory cascade activating IL-6 which results in the expression of VEGF and ICAM1, among other molecules. Regardless of the etiology of vascular pathology, all three disease entities (diabetic retinopathy, AMD, and RVO) share a common mechanism of resulting subretinal or intraretinal fluid accumulation and angiogenesis. Although VEGF has been identified as a key player and anti-VEGF injections are vital in the treatment, the utility of IL-6 inhibitors could be investigated as an additional therapeutic target based on the current understanding of the role of IL-6 in these diseases.

## 5. Conclusions

There is abundant experimental data that supports the vital role of IL-6 in the molecular pathophysiology of macular edema from a variety of etiologies. Patients with a long-standing history of refractory uveitic CME unresponsive to multiple first-line immunomodulatory therapies may benefit from the use of IL-6 inhibitors, including tocilizumab and sarilumab. Although their utility has been most documented in the treatment of NIU and NIU-related macular edema, the use of IL-6 inhibitors might be extended to inflammatory CME in the context of complications from cataract surgery, acute retinal necrosis, and autoimmune retinopathy. IL-6 remains an important player in triggering the innate inflammatory cascade and in activating VEGF. While there are no studies recorded on the use of IL-6 inhibitors in treating CME from retinal vascular conditions, such as diabetic retinopathy or RVO, IL-6 inhibitors may provide another treatment option due the role of IL-6 in vascular inflammation and disruption of the blood–retina barrier. By reviewing studies published on IL-6 and CME, we hope to highlight their role in treating resistant NIU-related macular edema and beyond.

## Figures and Tables

**Figure 1 ijms-24-04676-f001:**
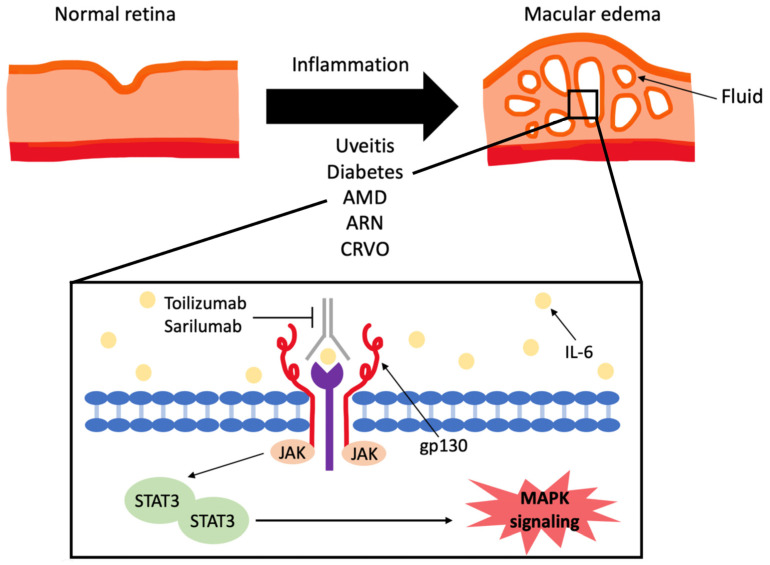
Interleukin-6 inhibitors in the treatment of macular edema. Abbreviations: age related macular degeneration (AMD), acute retinal necrosis (ARN), central retinal vein occlusion (CRVO), interleukin-6 (IL-6), interleukin-6 receptor (IL-6R), glycoprotein 130 (gp130), Janus activated kinase (JAK), signal transducer and activator of transcription 3 (STAT-3), JAK-SHP2-Ras-mitogen activated protein kinase (MAPK).

**Figure 2 ijms-24-04676-f002:**
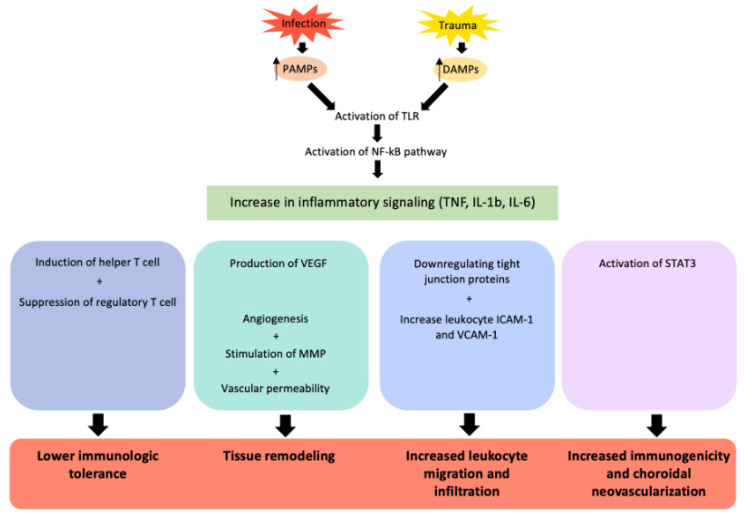
Diagram of IL-6 downstream signaling pathways. Abbreviations: pathogen-associated molecular patterns (PAMPs), damage-associated molecular patterns (DAMPs), Toll-like receptors (TLR), nuclear factor kappa B (NF-kB), tumor necrosis factor (TNF), interleukin-1b (IL-1b), interleukin-6 (IL-6), vascular endothelial growth factor (VEGF), intracellular adhesion molecule 1 (ICAM-1), vascular cell adhesion protein 1 (VCAM-1), signal transducer, and activator of transcription 3 (STAT-3).

## Data Availability

The data that support the findings of this study are available from the corresponding author, L.S., upon reasonable request.

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
