# Peer review of "Interleukin-6 and Macular Edema: A Review of Outcomes with Inhibition"

_ijms, 2023, doi:10.3390/ijms24054676_

Round 1

Reviewer 1 Report

1.       The title should be changed alignment with the text concentration  OR

2.       Both the introduction and the 2. The role of interleukin-6 in ocular pathology should be extended and more comprehensive. Although the manuscript title started with molecular mechanism of …, this means just consumes about 25-30% of the manuscript text, and the rest percentage goes for anti-IL-6 inhibitors?

3.       Authors recommend inserting a methods section which should include the last part of their introduction with more details, such as the keywords they used yielded how many papers, the number of the included and excluded papers in each stage of screening, etc..

4.       Your figure is very poor in the data presented (draw) and text. It is recommended to add more figures explaining more details of molecular pathophysiological effects of IL-6 in macular edema and where monoclonal inhibitors interact per disease stage development.

Author Response

  1. We have changed the title to align with the text concentration.
  2. We have chosen to change the title to align with text concentration as above.

  3. We had originally submitted the manuscript with a Methods section but the editor asked us to remove it to align with their format for reviews. We have added additional information regarding the key words and number of included and excluded papers into the current format.

  4. We have added an additional figure on the interaction of monoclonal inhibitors in the pathogenesis of macular edema. Unfortunately, macular edema does not have definable stages, therefore we can only demonstrate the general pathway involved without staging.

Reviewer 2 Report

The topic is very original. The paper only needs e review of english form to make the speech smoother and more flowing.

Author Response

We have edited the English throughout the text to make it smoother.

Reviewer 3 Report

This short review by Yang et al. summarizes the roles of IL6 in macular edema, and also describes the studies that have looked at the efficacy of IL6R antagonists in treating these diseases. The paper does a solid job covering these topics while remaining accessible for scientists who may be new to this particular field. I have no major comments.

Author Response

Thank you for the time taken to review our manuscript.

Round 2

Reviewer 1 Report

thank you for your reply, but I could not find Figure 2? please be sure you are uploading the correct version.